# Interactive Structural Analysis of KH3-4 Didomains of IGF2BPs with Preferred RNA Motif Having m^6^A Through Dynamics Simulation Studies

**DOI:** 10.3390/ijms252011118

**Published:** 2024-10-16

**Authors:** Muhammad Fakhar, Mehreen Gul, Wenjin Li

**Affiliations:** 1Institute for Advanced Study, Shenzhen University, Shenzhen 518060, China; mfakhar@bs.qau.edu.pk (M.F.); mehreengull@bs.qau.edu.pk (M.G.); 2College of Civil and Transportation Engineering, Shenzhen University, Shenzhen 518060, China

**Keywords:** m^6^A, IGF2BPs, KH4 domain, 503GKGG506 motif, molecular dynamics simulation

## Abstract

m^6^A modification is the most common internal modification of messenger RNA in eukaryotes, and the disorder of m^6^A can trigger cancer progression. The GGACU is considered the most frequent consensus sequence of target transcripts which have a GGAC m^6^A core motif. Newly identified m^6^A ‘readers’ insulin-like growth factor 2 mRNA-binding proteins modulate gene expression by binding to the m^6^A binding sites of target mRNAs, thereby affecting various cancer-related processes. The dynamic impact of the methylation at m^6^A within the GGAC motif on human IGF2BPs has not been investigated at the structural level. In this study, through in silico analysis, we mapped IGF2BPs binding sites for the GGm^6^AC RNA core motif of target mRNAs. Subsequent molecular dynamics simulation analysis at 400 ns revealed that only the KH4 domain of IGF2BP1, containing the 503GKGG506 motif and its periphery residues, was involved in the interaction with the GGm^6^AC backbone. Meanwhile, the methyl group of m^6^A is accommodated by a shallow hydrophobic cradle formed by hydrophobic residues. Interestingly, in IGF2BP2 and IGF2BP3 complexes, the RNA was observed to shift from the KH4 domain to the KH3 domain in the simulation at 400 ns, indicating a distinct dynamic behavior. This suggests a conformational stabilization upon binding, likely essential for the functional interactions involving the KH3-4 domains. These findings highlight the potential of targeting IGF2BPs’ interactions with m^6^A modifications for the development of novel oncological therapies.

## 1. Introduction

The methylation occurring at position N^6^ in adenines, referred to as m^6^A modification, is the most common internal modification found in messenger RNA (mRNA) [1,2]. Approximately one out of every two hundred adenine bases undergo this modification, playing a crucial role in key biological processes. As a result, m^6^A modification significantly contributes to fundamental events in the field of biology [1,3,4]. At the molecular level, the incorporation of m^6^A modifies RNA structures, impacting their capacity to engage in protein–RNA interactions [5]. Consequently, this modification plays a regulatory role in cellular processes such as RNA processing [6], translation [7,8], and transcript stability [9]. In turn, malfunctions in the cellular machinery that regulates m^6^A modification have been associated with pathologies such as obesity [10], cancer [11], and neurodegeneration [12]. The GGACU is the most common consensus sequence in target transcripts, with a GGAC m^6^A core motif positioned in the 3′-untranslated regions (UTRs), coding sequences around stop codon regions, and 5′-UTRs, and >80% of the targets contain at least one m^6^A site [1,13].

The methylation process of m^6^A is governed by regulators categorized as ‘writers,’ ‘erasers’, and ‘readers’ [14]. Methyltransferases, known as ‘writers’, add m^6^A methyl groups to RNA, whereas demethylases, referred to as ‘erasers’, reversibly erase RNA m^6^A methylation. ‘Readers’, consisting of proteins, execute the biological functions associated with m^6^A methylation [15]. Among the recently known members of this reader group are IGF2BPs, including IGF2BP1, IGF2BP2, and IGF2BP3 [16]. IGF2BPs, conserved single-stranded RNA-binding proteins, showcase a structural organization with six canonical RNA-binding domains, including two RNA recognition motif (RRM) domains and four K homology (KH) domains [17]. The amino acid sequences of these IGF2BPs have been confirmed to share a similarity of over 56% [18]. Notably, IGF2BP1 and IGF2BP3 show the highest sequence identity, reaching 73% [17,18]. This significant sequence similarity may suggest that IGF2BPs perform similar biochemical functions, such as RNA binding, regardless of the organism, tissue, or cell type [13]. Recent studies have established that differences in RNA specificity between IGF2BP1 and IGF2BP2 in the KH3-4 domains are due to differences in the amino acids within the variable loops. Although their findings firmly imply the co-evolution of interface amino acids and RNA nucleotides for target specificity, the structural basis for this remains unclear [19,20]. The KH domains within IGF2BPs, specifically the KH3-4 domains, have been verified as the elements accountable for recognizing and binding to m^6^A [16]. These KH domains have the ability to identify short motifs consisting of 3–6 bases through their GXXG motifs situated in a distinctive flexible loop [21,22]. These domains are frequently found in several copies, acting cooperatively or independently. This arrangement serves to enhance the affinity and/or specificity of KH domains for their targets [23,24,25,26].

IGF2BPs have a preference for binding to the “UGGAC” consensus sequence containing the “GGAC” m^6^A core motif. This binding at the m^6^A methylation modification site in MYC mRNA enhances both the stability and translation efficiency of MYC mRNA in HCC and cervical cancer [16]. The proposed experimental binding mode between gallus gallus IGF2BP1 KH3-4 domains and the RNA target UCGGm^6^ACU containing the GGm^6^AC motif is well understood (PDB ID: 8COO), but the dynamic impact of m^6^A methylation modification on the KH3-4 domains of human IGF2BP1-3 homologous aliases containing this motif has not been investigated at the structural level. Therefore, through in silico analysis, we mapped IGF2BPs binding regions for the GGm^6^AC RNA core motif of target mRNAs to modulate gene expression in cancers and further elucidated the comparative dynamics, binding mode, stability, displacement, and subtle structural changes of these complexes.

## 2. Results

### 2.1. Modeling and Comparative Analysis of KH3-4 Domains of IGF2BPs

The KH domains of IGF2BPs are very important for binding to RNA [16]. Especially, it has been observed that KH3-4 domains have a role in recognizing and binding to RNA motifs which have an m^6^A modification [16,27]. In humans, each KH domain of IGF2BPs shows a higher sequence identity to the corresponding KH domain in another IPM paralog than to other KH domains within the same protein [20]. The experimental 3D structure of gallus gallus IGF2BP1 with mutated KH3-4 domains (KK422-423DD) bound to the RNA pattern “UCGGm^6^ACU” containing the GGAC motif is well understood (PDB ID: 8COO). These mutations particularly occur in the KH3 domain of gallus gallus IGF2BP1. We observed that gallus gallus KH3-4 domains of IGF2BP1 share 96.8%, 82.2%, and 84.5% sequence identity with the KH3-4 domains of human IGF2BP1, IGF2BP2, and IGF2BP3, respectively. We discovered that the sequence numbering varies across all IGF2BP proteins, despite having conserved residue identities at specific positions. To ensure consistency in our results, we used the KH3-4 domains of gallus gallus as a reference and adjusted the numbering of the KH3-4 domain sequences in all human IGF2BPs accordingly (Figure 1A). The experimental 3D structurae information of KH3-4 domains of IGF2BPs with RNA motifs is mentioned in Appendix A. All KH3-4 domains contain six α helices and six β sheets (Figure 1A). The KH3 domain in all IGF2BPs ranges from 404 to 469 amino acids, having an important 421GXXG424 loop which is present between α1 and α2. Domain KH4 of all IGF2BPs has a sequence from 486 to 563 amino acids and it keeps a conserved 503GKGG506 loop between α4 and α5 which basically recognizes a short motif of 3–6 nucleotides (nt) [21,22]. The KH3 and KH4 domains are interconnected through a short sequence of amino acids (470–487 AA) called linker (Figure 1B). The human KH3-4 domains of all IGF2BPs were superimposed with gallus gallus KH3-4 domain via UCSF Chimera version 1.17.3, and the resultant RMSD value of 0.506 Å suggested the conservation and reliability of all structures (Figure 1B). For reader convenience, the KH3-4 domains of gallus gallus are labeled with the short code “GG1”, while the KH3-4 domains of all human IGF2BPs are referred to with the codes “Hu1”, “Hu2”, and “Hu3” throughout the manuscript (Figure 1).

### 2.2. Molecular Docking Analysis

A molecular docking study of the KH3-4 domains was conducted using a double mutation (DD) in the KH3 motif’s GXXG sequence of human IGF2BP1-3 and gallus gallus IGF2BP1. This analysis focused on the interaction with the critical RNA core motif GGAC containing m^6^A. Docking was performed utilizing Alphafold3 [28], along with the GRAMM [29] and HDock [30,31] web servers. The docked complexes of all KH3-4 domains of IGF2BPs with GGm^6^AC were evaluated on the basis of the highest interface predicted template modeling (ipTM) score, their binding energy value, and the binding site. AF3-specific predicted ipTM scores for KH3-4 domains of human IGF2BP1, IGF2BP2, IGF2BP3, and gallus gallus IGF2BP1 with GGm^6^AC core motif of the RNA were 0.93, 0.92, 0.90, and 0.93, respectively. HDock and GRAMM-specific predicted energy values are mentioned in Appendix A. The binding pattern of all human KH3-4 domains of IGF2BPs “Hu1”, “Hu2”, “Hu3”, and gallus gallus KH3-4 domains of IGF2BP1 “GG1” with GGm^6^AC was quite similar to that reported for the gallus gallus IGF2BP1 (KH3-4)-UCGGm^6^ACU complex (PDB ID: 8COO) (Appendix A). The binding was observed in the conserved 503GKGG506 loop (olive green) and its periphery hydrophobic groove (yellow) of the KH4 domain of all respective IGF2BPs (Figure 2). Interactions of GGm^6^AC in complex with KH3-4 domains are listed in Appendix A.

### 2.3. Molecular Dynamics Simulation Analysis

All IGF2BPs apo KH3-4 domains and their KH3-4-GGm^6^AC complexes were further estimated through MD simulation analyses to examine the dynamic behavior, stability, and structural changes of KH3-4 domains upon GGm^6^AC binding, initially over 100 ns and 200 ns, and subsequently extended to 400 ns. Appendix A presents the RMSD (root mean square deviation) analysis for the bound states across three different MD simulation time scales: 100 ns, 200 ns, and 400 ns. The secondary structure elements stability, binding mode extension, and conformational changes of the simulated complexes were assessed by plotting the RMSD, Rg (radius of gyration), SASA (solvent accessible surface area), RMSF (root mean square fluctuation), hydrogen bonding, PCA (principal component analysis) and binding energy. RMSD analysis suggested an overall convergence pattern and equilibration for each complex. For reader convenience, the apo KH3-4 domains of gallus gallus IGF2BP1 and human IGF2BP1-3 are labeled with the short codes “Apo_GG1”, “Apo_Hu1”, “Apo_Hu2” and “Apo_Hu3”, respectively. Their bound states with GGm^6^AC are represented by the codes “GG1_bound”, “Hu1_bound”, “Hu2_bound” and “Hu3_bound” throughout the manuscript. Apo-KH3-4 domains of IGF2BPs were utilized as references to measure the RMSD for each complex throughout the 400 ns time scale. The backbone RMSD profile for GG1_bound, Hu1_bound, Hu2_bound, and Hu3_bound complexes showed stability throughout the time scale as compared to their apo states. During MD simulation runs, the average RMSD values for the backbone of GG1_bound, Hu1_bound, Hu2_bound, and Hu3_bound complexes were below 0.238 nm, 0.118 nm, 0.163 nm, and 0.156 nm, respectively, suggesting stability of the systems (Figure 3). The overall RMSD profile analysis for all complexes revealed that all systems are well equilibrated (Figure 3).

We also compared the bound states of IGF2BPs (KH3-4) to observe the comparative stability of these complexes with each other. In GG1_bound complexes, we observed RMSD fluctuation between approximately 0.1 nm and 0.43 nm, whereas in Hu1_bound the RMSD remained more stable, ranging between ~0.05 nm and ~0.15 nm. In Hu2_bound RMSD ranges between ~0.1 nm and ~0.3 nm, and Hu3_bound noticeably displayed an RMSD fluctuation between ~0.1 nm and ~0.25 nm (Figure 4A). In summary, the RMSD across all complexes generally falls within the range of ~0.1 nm to ~0.43 nm, with Hu1_bound being the most stable and GG1_bound showing the largest fluctuations. The compactness of the reader protein-docked complexes has been assessed via the computation of the radius of gyration (Rg). The result confirmed that the Rg values of the GG1_bound, Hu1_bound, Hu2_bound, and Hu3_bound stayed steady with a range between ~1.58 nm to ~1.75 nm right through the MD simulation time frame of 400 ns (Figure 4B). Additionally, we also performed a SASA analysis to understand the solvent behavior of all complexes. SASA profiles revealed that the docked complexes had average values with a range of ~90 nm^2^ to ~109 nm^2^ (Figure 4C). The average SASA values of GG1_bound, Hu1_bound, Hu2_bound, and Hu3_bound were 103.723 nm^2^, 95.774 nm^2^, 98.929 nm^2^, and 93.214 nm^2^, respectively.

RMSF values estimated the magnitude of residual fluctuations to compare the structural changes among apo-KH3-4 domains of IGF2BPs and its bound forms with GGm^6^AC. As revealed in Figure 5A,B, the residues involved in binding to RNA motif in bound states such as GG1_bound and Hu1_bound showed similar fluctuations, which indicated that GGm^6^AC motifs could form an interaction with amino acids in the active pocket including the 503GKGG506 loop and its nearby potent residues of the KH4 domain of IGF2BPs. Meanwhile, in plots C (Hu2_bound) and D (Hu3_bound), we observed minor fluctuation in the 503GKGG506 loop of the KH4 domain as compared to their relevant apo states, but interestingly, despite these relatively stable fluctuations, the RNA motif appears to shift away from the KH4 domain binding region towards the KH3 domain. Specifically, the RNA seems to be reorienting towards nearby RNA-binding motifs, such as the 421GXXG424 sequence present in the KH3 domain in Hu2_bound and Hu3_bound. We observed that particularly loop regions, the 421GXXG424 motif in KH3, the linker region between KH3-4 domains (471–485 AA), and the 503GKGG506 motif with periphery residues in the KH4 domain in all apo systems displayed greater fluctuations. The structural superposition of IGF2BPs (KH3-4)-GGm^6^AC complexes with their apo stats at the 400 ns MD simulation time scale is represented in Figure 5E–H. After the superimposition, the binding region including the 503GKGG506 loop displayed displacement changes in apo as compared to bound states of all IGF2BPs KH3-4 domains (Figure 5E–H). Especially the Arg524 residue in bound complexes such as GG1_bound and Hu1_bound (Figure 5E,F) got closer to the RNA motif. We also measured the RMSD values by the superimposition of apo IGF2BPs (KH3-4) structures and their bound states with RNA motifs at 400 ns. For apo_GG1 and its GG1_bound complex (Figure 5E), the general RMSD value was 1.059 Å, and the active pocket RMSD value was 1.159 Å. For the apo_Hu1 and its bound complex (Figure 5F), the general and active pocket RMSD values were 0.932 Å and 0.754 Å, respectively. For the apo and Hu2_bound complex (Figure 5G), the general and active pocket RMSD values were 1.007 Å and 1.066 Å, respectively. For the apo_Hu3 and Hu3_bound complex (Figure 5H), the general and active pocket RMSD values were 0.94 Å and 0.826 Å, respectively. Overall, comparative analysis of the IGF2BPs (KH3-4) in the apo and bound states, based on structural superimposition, revealed that gallus gallus IGF2BP1 (KH3-4) exhibited slightly higher RMSD values compared to the other systems. We mostly observed displacement changes in the binding site and different loop regions of IGF2BPs (KH3-4) (Figure 5). RMSF comparative analysis of bound IGF2BPs (KH3-4) complexes showed that fluctuation was only observed in loop regions of all complexes with nearby similar patterns and that the binding region remained less fluctuated (Figure 6A).

All IGF2BPs (KH3-4)-GGm^6^AC complexes were also examined for hydrogen bonding shifts. The average number of hydrogen bonds formed over time for different bound states is shown in Figure 6B. The GG1_bound complex shows an average of 4.631 hydrogen bonds, whereas Hu1_bound, Hu2_bound, and Hu3_bound complexes have 4.539, 1.966, and 2.4 hydrogen bonds, respectively, at the 400 ns MD simulation time scale. The average number of hydrogen bonds fluctuated over time for each bound state. In GG1_bound (pink) and Hu1_bound (green), the RNA motif generally showed a higher number of hydrogen bonds, reflecting more stable binding interactions compared to the other states at 400 ns. In contrast, Hu2_bound and Hu3_bound exhibited a marked reduction in the number of hydrogen bonds over time, particularly in the latter stages of the simulation (after 200 ns). This suggests that as the RNA motif moved away from the KH4 domain, the loss of hydrogen bonding contributed to decreasing binding stability (Figure 6B). The hydrogen-bonding pattern implied stable interactions in harmony with the RMSD distribution. Overall, this figure provides insights into the flexibility and stability of the protein structure in different bound states by analyzing RMSF and hydrogen bond formation.

To illustrate the binding pattern and orientation shifts, dynamic trajectories of all IGF2BPs KH3-4 domains with GGm^6^AC were generated at different time scales: 0, 50, 100, 150, 200, 250, 300, 350, and 400 ns time intervals. We observed that there is stability and improvement of the interaction pattern in GG1_bound and Hu1_bound at 400 ns in the MD simulation, in which the RNA motif remained in binding sites such as the 503GKGG506 loop and its periphery potent residues of the KH4 domain. In the GG1_bound complex, Gln526 formed a hydrogen bond with G_1_, and residues Lys504 and Gly505 showed a hydrogen bond with guanine base G_2_. Additionally, the cytosine C_4_ base moved closer to the binding loop 503GKGG506 and formed hydrogen bonds with nearby residues such as Thr508 and Asn510. Several other residues, including Ala498, Gly499, Ile502, Gly503, Gly506, Lys507, Thr508, Val522, Pro523, Arg524, and Pro528, contributed to the overall stability of the complex through hydrophobic interactions with the RNA motif at 400 ns in the MD simulation (Figure 7A). The interaction between GGm^6^AC and human IGF2BP1 (KH3-4) “Hu1” complex showed that m^6^A is surrounded by a hydrophobic groove consisting of the 503GKGG506 loop and its potent periphery residues like Val521, Val522, and Pro523 (Figure 7B). The G_1_ nucleotide showed hydrogen bonding with Asp525 and Gln526 whereas the G_2_ base exhibited hydrogen bonding with Arg524 in the Hu1_bound complex. The other important residues, such as Gly499, Arg500, Ile502, Gly503, Lys504, Gly505, Gly506, Val522, Pro523, and Val534, were implicated in hydrophobic interactions in this complex (Figure 7B) at 400 ns in the MD simulation. In Hu2_bound a large number of potent binding site residues, such as Ser495, Ala498, Ile502, Gly503, Lys504, Gly505, Gly506, Val520, Ile521, Val522, and Pro523, were involved in hydrophobic interactions with GGm^6^AC (Figure 7C) whereas Gln526 and Asp525 were involved in hydrogen bonding with G_1_ and G_2_ nucleotide bases of the RNA motif at 150 ns. In the Hu3_bound complex RNA motif GGm^6^AC presented the main interaction with Ser495, Phe496, Ala498, Gly499, Ile502, Gly503, Lys504, Gly506, Val521, Val522, Pro523, Arg524, Gln526, Val534, and Gly505 in human IGF2BP3 (KH3-4) (Figure 7D) at 120 ns in the MD simulation. In all complexes, the binding residues at different time scales are mentioned in Appendix A. Overall, the Hu2_bound and Hu3_bound complexes demonstrated that they were primarily bound to the KH4 domain but lost stability after shorter simulation times. The Hu2_bound complex stayed stable for nearly 150 ns before the RNA motif detached from the KH4 domain and moved toward the KH3 domain. Likewise, in Hu3_bound, the RNA motif stayed bound to the KH4 domain for 120 ns, after which it transitioned towards the KH3 domain (Appendix A).

### 2.4. Principal Component Analysis

To investigate the conformational changes of IGF2BPs (KH3-4) upon GGm^6^AC RNA motif binding, we performed principal component analysis (PCA) on proteins in both apo (unbound) and bound states. PCA is a statistical technique that simplifies MD trajectories by isolating the overall movement of backbone atoms while retaining much of the other changes. It computes the covariance matrix of positional variations in backbone atoms, which can reveal the changing aspects and coordinated movements of the complexes. The plot, shown in Figure 8, is the 2D projection of the trajectories for the two major principal components, PC1 and PC2, for gallus gallus apo_GG1 and its bound state, as well as human apo_Hu1and its bound state. The plots revealed diverse conformational distributions in the 2D space across these systems. In apo_GG1 (panel A), PC1 accounted for 62.09% of the variance, while in GG1_bound (panel B), PC1 explained 64.79% of the variance. Notably, apo_GG1 demonstrated a more spread-out distribution of points along PC1 compared to GG1_bound (Figure 8A,B). Plots C (apo_Hu1) and D (Hu1_bound) are also PC1 vs. PC2 PCA plots, with PC1 accounting for 56.91% and 60.09% of the variance, respectively. The distribution of points in plot D (bound state) is more compact, particularly along the PC1 axis, indicating that the conformational landscape is more restricted when the RNA motif is bound to the protein. On the other hand, bound states (plots B and D) showed tighter clustering or less variance along PC1, meaning that these systems are less dynamic compared to the apo states. The apo states exhibited high conformational flexibility, as indicated by the broad spread of projections on the principal components. In contrast, the bound states showed more confined distributions, reflecting a stabilizing effect upon GGm^6^AC binding, which strongly correlates with the previous analyses such as RMSD, RMSF, SASA, and others. These results suggest that GGm^6^AC binding induces structural stabilization in the protein, which may be crucial for its functional regulation. Appendix A presents the k-means clustering analysis (K = 3) applied to the principal component analysis (PCA) data of the apo and bound complexes, where the x-axis represents PC1 and the y-axis represents PC2. Each subplot (A–D) shows the distribution of frames colored by cluster and the corresponding centroids (marked by red crosses). The k-means clustering results suggested that both the apo and bound complexes exhibit distinct conformational states, represented by the three clusters.

For the analysis of the energy landscape, 400 ns simulated trajectories were subjected to MM/PBSA to estimate the binding impacts of individual residues. The KH3-4 domains of IGF2BPs, such as GG1 and Hu1, with GGm^6^AC RNA motif, displayed total binding energy values of −78.494 and −258.627 KJ/mol, respectively, as shown in Table 1. Van der Waals (E_vdw_) interactions, and electrostatic (E_elec_) and nonpolar solvation energies (ΔG_sol-nonpolar_) contributed negatively to the total binding energy, whereas polar solvation energy (ΔG_sol-polar_) contributed positively (ΔG_binding_).

Our results highlighted the dominant role of electrostatic interactions in stabilizing the association between GGm^6^AC and the KH3-4 domains of IGF2BP1. The decomposition analysis of the binding enthalpy revealed multiple contributions from individual residues (Figure 9), implying a stable interaction pattern between the KH3-4 domains of gallus gallus and human IGF2BP1 with GGm^6^AC. Arg500, Gly503, Lys504, Gly506, Lys507, Val522, Pro523, and Arg524 residues showed predominant energy contributions during GGm^6^AC RNA motif binding (Figure 9). These key residues identified across all complexes are critical for the enthalpy-driven stabilization of the interactions. These findings were consistent with the RMSF analysis, which showed that these residues remained stable throughout the simulation run (Figure 5A,B). The negative binding energy values for GG1 and Hu1 in complex with RNA motif GGm^6^AC indicated higher binding affinities (Table 1). The magnitude of the negative values reflects the strength of these interactions, with Hu1_bound being the most stable as compared to GG1_bound.

## 3. Discussion

m^6^A methylation is the most frequent modification of mRNA in eukaryotes [32], and an increasing number of studies reveal that disorders of m^6^A can trigger cancer progression [33]. The m^6^A modification refers to a methyl group attached to the N^6^ position of adenosine and mostly occurs in the common sequence DRACH (D = G/A/U, R = G/A, and H = A/C/U) [34]. The GGACU sequence is a prevalent consensus motif in target transcripts, with m^6^A sites commonly found in 3′-UTRs, coding sequences near stop codons, and 5′-UTRs [13,35]. The methylation process of m^6^A is modulated by regulators such as ‘writers’, ‘erasers’, and ‘readers’, as shown in Figure 10. We have focused on newly identified m^6^A ‘readers’ insulin-like growth factor 2 mRNA-binding proteins (IGF2BPs), which regulate gene expression by binding to m^6^A sites on target mRNAs influencing cancer stem cells, proliferation, migration, glycolysis, cell cycles, angiogenesis, and therapy resistance [16,36,37,38,39,40,41].

In our study, the docking analysis revealed that the GGm^6^AC motif interacted with all IGF2BPs: gallus gallus GG1 and human Hu1, Hu2, and Hu3, specifically engaging the 503GKGG506 motif and its periphery residues within the KH4 domain. However, at 400 ns in MD simulations, the GGm^6^AC RNA motif maintained the interaction with the KH4 domain of GG1 and Hu1, while in Hu2 and Hu3, the RNA motif shifted away from KH4 and moved closer to the KH3 binding loop. This observation supports earlier findings showing that the 421GXXG424 motif in the KH3 domain and the 503GKGG506 motif in the KH4 domain are vital for the recognition and binding of m^6^A-modified RNA [16,20,42]. However, the dynamic behavior of these complexes at the structure level has not been previously investigated. Notably, earlier studies have also shown that the IMP1 KH4 domain recognizes a GGAC RAC-like sequence [43,44], suggesting that this domain is most likely accountable for facilitating IGF2BP1/IMP1 recognition of m^6^A methylated RNAs [45]. Molecular dynamics (MD) simulation analysis at 400 ns showed an interaction between the GGm^6^AC RNA motif and KH4 domain’s potent loop 503GKGG506 in which the Lys504 sidechain is involved in a specific contact [20], as well as its nearby hydrophobic groove consisting of residues like Val521, Val522 and Pro523 [45], in GG1_bound and Hu1 complexes (Figure 7A,B). Notably, we detected conservation of the 520VVVP523 sequence in the binding region (500–530 AA) of both GG1 and Hu1. However, in Hu2, this sequence is 520VIVP523, with Isoleucine substituting Valine, potentially affecting RNA binding on the KH4 domain at 400 ns in the MD simulation. In previous work, it was established that mutating two crucial residues in the hydrophobic cradle (V522I/P523S) of the KH4 domain disrupted m^6^A-RNA specificity by altering the hydrophobic interaction with the RNA methyl group. This double mutation reversed m^6^A recognition while maintaining the overall structure, confirming the critical role of these residues in m^6^A binding [45]. Biswas et al. (2019b) recently explained that differences in RNA specificity between the KH3-4 domains of IGF2BP1 and IGF2BP2 are verified by differences in the amino acids present in the variable loops [19]. This suggests a conformational stabilization upon binding, likely essential for the functional interactions involving the KH3-4 domains. Across all panels (A–D) in Figure 8, a common trend is observed where IGF2BPs exhibit significant flexibility in the apo (unbound) state, as evidenced by the broader spread of projections. In contrast, the bound states consistently show more confined distributions, indicating that GGm^6^AC binding stabilizes the protein structure. This stabilization likely results in a more defined conformation, reducing the structural variability that is characteristic of the unbound form. The PCA clearly illustrates the impact of binding on the structural dynamics of the protein. Furthermore, the binding energies (ΔG) calculated using the MM/PBSA method for the complexes GG1 and Hu1 with GGm^6^AC were all negative, indicating that each complex has favorable interactions and is likely to be stable. Negative values of binding energy suggest that the formation of these complexes is thermodynamically favorable, as the interaction between the molecules is energetically beneficial [46]. The analysis of specific residues provides valuable insights into the molecular details of RNA–protein interactions. The residues Arg500, Gly503, Lys504, Gly506, Lys507, Val522, Pro523, and Arg524 are repeatedly observed to contribute significantly to the binding energy in all complexes. These residues are likely key interaction points where the protein interfaces with the RNA motif GGm^6^AC [45].

## 4. Material and Methods

### 4.1. Data Set

Insulin-like growth factor-2 mRNA-binding proteins (IGF2BP1–3) are members of a conserved family of single-stranded RNA-binding proteins [17]. The mutated KH3-4 domains of IGF2BPs were studied instead of the wild-type because most available experimental data is for the mutant form, allowing for better comparison with the computational results. The experimental mutated (KK422–423DD) 3D structure of gallus gallus KH3-4 domains of IGF2BP1 (PDB ID: 8COO) in complex with the RNA target UCGGm^6^ACU is well understood. Sequences of the KH3-4 domains of human IGF2BP1 (ID: Q9NZI8) (403–560 AA), IGF2BP2 (ID: Q9Y6M1) (403–563 AA), IGF2BP3 (ID: O00425), (404–552 AA) and gallus gallus IGF2BP1 (ID: O42254) (404–567 AA) were isolated through uniportKB [47], and subjected to multiple sequence alignment. Clustal W www.genome.jp/tools-bin/clustalw (accessed on 25 March 2024) is extensively used for MSA analysis.

### 4.2. Molecular Docking

KH domains, especially the KH3-4 didomains, are critical for the binding of IGF2BPs to m^6^A-modified RNAs [18,32,44]. In this study, the KH3-4 domains of human IGF2BP1-3 and gallus gallus IGF2BP1, with the KH3 RNA-binding capability knocked out via a double DD mutation, were primarily docked with the consensus GGACU RNA motif using AlphaFold3 [28]. AF3 generated five docking models of the KH3-4 domains binding to the GGACU RNA motif, which were subsequently utilized for structural analysis. The top-ranking complexes were selected based on the highest inter-protein template modeling (ipTM) scores. Following selection, the Uracil nucleotide was cleaved from the complexes. In the selected complexes, the GGAC RNA motif was modified to include N^6^-methyladenosine (m^6^A) at the adenine position through CHARMM-GUI [48]. The resulting m^6^A-modified complexes were subsequently prepared for molecular dynamics (MD) simulations using CHARMM-GUI, with all necessary parameters and force fields generated to accurately represent the m^6^A modifications. We further validated our docking results between the KH3-4 domains of IGF2BPs and GGm^6^AC using GRAMM [29] and HDock [30,31] docking web servers.

### 4.3. Molecular Dynamics Simulation Analysis

To achieve deeper insight and assess dynamic behavior, conformational adjustments, and interaction steadiness, molecular dynamics (MD) simulations were initially conducted for all complexes, both in their apo and RNA-bound forms, for 100 ns. These simulations were subsequently extended to 200 ns and conclusively to 400 ns for each complex, achieving comprehensive sampling of the system dynamics. All MD simulations of apo IGF2BPs (KH3-4) domains and bound complexes of IGF2BPs (KH3-4) domains with the GGm^6^AC core motif of RNA were accomplished through GROMACS 2022.5 on a supercomputer [49,50], using a CHARMM36 force field [48,51,52,53]. The systems for the MD runs were constructed using the CHARMM-GUI input generator [48]. In all IGF2BPs bound complexes with GGAC RNA motifs, adenosine was methylated at the N^6^ nitrogen atom (i.e., N^6^-methyladenosine (m^6^A) formation) through the CHARMM program [48]. The complexes were placed in a periodic rectangular water box employing the TIP3P water model [54,55], maintaining a 10 Å distance between the solutes and the box edges. All complexes were firstly centered in the periodic rectangular boxes with the following dimensions: human Apo-IGF2BP1(KH3-4), 8.0 × 7.54 × 6.53 nm; IGF2BP1(KH3-4)-GGm^6^AC complex, 8.0 × 7.54 × 6.53 nm; Apo-IGF2BP2(KH3-4), 8.10 × 7.63 × 6.61 nm; IGF2BP2(KH3-4)-GGm^6^AC complex, 8.10 × 7.63 × 6.61 nm; Apo-IGF2BP3(KH3-4), 8.0 × 7.54 × 6.53 nm; IGF2BP3(KH3-4)-GGm^6^AC complex, 7.90 × 7.44 × 6.45 nm; and gallus gallus Apo-IGF2BP1(KH3-4), 8.40 × 7.91 × 6.85 nm; and IGF2BP1(KH3-4)-GGm^6^AC complex, 8.30 × 7.82 × 6.77 nm). Subsequently, the systems were neutralized by adding an appropriate number of Na+ and Cl− ions through the Monte-Carlo ion placing method, simulating a salt solution concentration of 0.15 M. Molecular dynamics simulations were conducted at a constant pressure of 1 atm and a temperature of 303 K, employing a Nosé–Hoover thermostat [56]. The simulation systems underwent an initial minimization process consisting of 5000 steps as follows: initially, four energy minimization steps were conducted with position restraints applied to the heavy atoms, using spring constants of 500, 250, 100, and 50 kcal (mol Å^2^)^−1^, respectively. Then, the resulting system was minimized without position restraints. This aimed to alleviate bad contacts and address poor geometry within the system. The LINCS algorithm [57] was used to constrain bonds involving hydrogen atoms through an integration time step of 2 fs. Long-range electrostatic interactions were computed with a 1 nm cut-off, using direct interaction via the efficient and smooth particle-mesh Ewald (PME) summation method [58]. The minimized structure was heated to 303 K and equilibrated in an NVT condition (constant volume and temperature). Further, 1 ns MD simulation under NPT ensemble [59] was initially conducted with position restraints applied to all heavy atoms with a force constant of 500 kcal (mol Å^2^)^−1^ using an isotropic Parrinello–Rahman barostat [60,61]. The position restraints were gradually released over five steps of 500 ps NPT simulations, with force constants applied to the heavy atoms being reduced progressively to 250, 100, 50, 10, and 5 kcal (mol Å^2^)^−1^. Then, a 500 ps simulation in the NPT ensemble with position restraints force constant of 5 kcal (mol Å^2^)^−1^ on backbone atoms was performed. Lastly, with the starting structure taken from the final 500 ps simulation, 400 ns production runs were carried out at the NPT ensemble for every system. PDB files were generated at 50 ns intervals to assess system stability and structural changes. The examination of MD trajectories involved the use of UCSF Chimera 1.17.3 and GROMACS software. The dynamic behavior and solidity of each system were analyzed via GROMACS modules, including g_rmsf, g_rms, g_gyrate, and g_hbond. Solvent-accessible surface area (SASA) was examined to understand the structural properties and interaction nature of biomolecules. SASA represents the area traced by a solvent molecule as it rolls around the molecular complex, coming into contact with the van der Waals surfaces of the atoms. The area is computed by Equation [62,63].
SASA=∑ⅈ (R/R2−zi2).D.Li

The value of *R* is calculated by addingthe radius of the solvent molecule to the van der Waals radius of the atom. *Z*_*i*_ represents the distance from the center of the sphere to section *i*; *D* is the distance between two consecutive sections, specifically between the *i*th and (*i* + 1)th sections; and *L*_*i*_ denotes the arc length computed on the section *i*.

### 4.4. Principal Component Analysis

Principal components analysis (PCA) or essential dynamics (ED) have been utilized to reduce the dimensionality of molecular dynamics (MD) simulations data, helping to determine the configuration space of inharmonic motion involving only a few degrees of freedom [64,65]. PCA is a method for analyzing the MD trajectory and determining the dominant modes in the overall molecular motion. The primary eigenvector projection in Cartesian trajectory coordinates was employed to detect the movement of structures within a multidimensional space [66]. In the ED analysis, we constructed a covariance matrix from the whole 400 ns simulation trajectories of the backbone Cα-atoms for both apo and bound complexes of IGF2BPs after eliminating rotational and translational motions. Furthermore, we calculated the eigenvectors and eigenvalues of the covariance matrices, along with the projections of the first two principal components. This PCA was performed via the built-in utilities of GROMACS, specifically gmx_covar and gmx_anaeig.

### 4.5. Binding Energy Calculation

The binding energy of the system was estimated using the Poisson–Boltzmann or generalized Born and surface area continuum solvation (MM/PBSA) method [67]. This approach improves docking energy by accounting for protein flexibility and offers a comprehensive energy composition. Equation (1) is applied to determine the binding energy of ligand–protein complexes.
ΔG_binding_ = G_complex_ − (G_protein_ + G_ligand_) (1)

G_protein_ and G_ligand_ correspond to the total energies of the isolated protein and ligand in the solvent, respectively, while G_complex_ indicates the total energy of the protein–ligand complex. The energies for each individual component, G_complex_, G_protein_, and G_ligand_, were calculated using the following equation:G_X_ = (E_MM_) + G_solvation_
(2)

Here, X denotes the protein–ligand complex. G_solvation_ is the energy of solvation, and E_MM_ is the molecular mechanics energy. The molecular mechanics potential energy was determined in a vacuum as follows:E_MM_ = E_bonded_ + E_non-bonded_ = E_bonded_ + (E_vdw_ + E_elec_) (3)

E_bonded_ contains bonded interactions such as bond, angle, dihedral, and improper interactions, and E_non-bonded_ holds non-bonded interactions, including electrostatic (E_elec_) and van der Waals (E_vdw_) interactions. The solvation energy (G_solvation_) is computed as the sum of the electrostatic solvation energy (G_polar_) and the apolar solvation energy (G_non-polar_).
G_solvation_ = G_polar_ + G_non-polar_
(4)

G_polar_ was estimated using the Poisson-Boltzmann (PB) Equation [68], and G_non-polar_ was computed using a solvent-accessible surface area (SASA) as follows:G_non-polar_ = γSASA + b (5)

Here, γ is a coefficient related to the surface tension of the solvent, and b is the fitting parameter.

## 5. Conclusions

This study provides a vital understanding of the molecular dynamics and structural interactions of IGF2BPs with the GGm^6^AC RNA core motif, with emphasis on the K3-4 domains. The results accentuate the key role of these domains in recognizing and binding m^6^A-modified RNA, which is vital for regulating RNA stability and gene expression. IGF2BP1 has showed substantially higher binding stability to the m^6^A-modified GGAC motif compared to IGF2BP2 and IGF2BP3, driven by conserved residues like Lys504, Gly506, and Val522 in the KH4 domain. This suggests that IGF2BP1 may play a more prominent role in post-transcriptional regulation, particularly in cancer pathways where m^6^A dysregulation is implicated. Remarkably, molecular dynamics simulations uncovered distinctive behaviors across IGF2BPs, with the RNA motif shifting from the KH4 to the KH3 domain in IGF2BP2 and IGF2BP3, suggesting functional divergence among the paralogs. Binding energy calculations using MM/PBSA further strengthened the stable interactions of IGF2BP1, revealing a thermodynamic preference for the KH4 domain’s hydrophobic environment to accommodate m^6^A. In contrast, IGF2BP2 and IGF2BP3 showed reduced binding stability, likely due to variations in key RNA-binding residues. Principal component analysis (PCA) further revealed that RNA binding induces structural stabilization, limiting the conformational flexibility of the KH3-4 domains, a key feature for efficient RNA-protein interactions. These findings strengthen the understanding of m^6^A-mediated RNA regulation by IGF2BPs and suggest IGF2BP1 as a promising therapeutic target for cancer treatment. These findings pave the way for precision therapies that modulate RNA-protein interactions, providing a strategic basis for future cancer treatments that specifically inhibit or enhance m^6^A recognition.

## Figures and Tables

**Figure 1 ijms-25-11118-f001:**
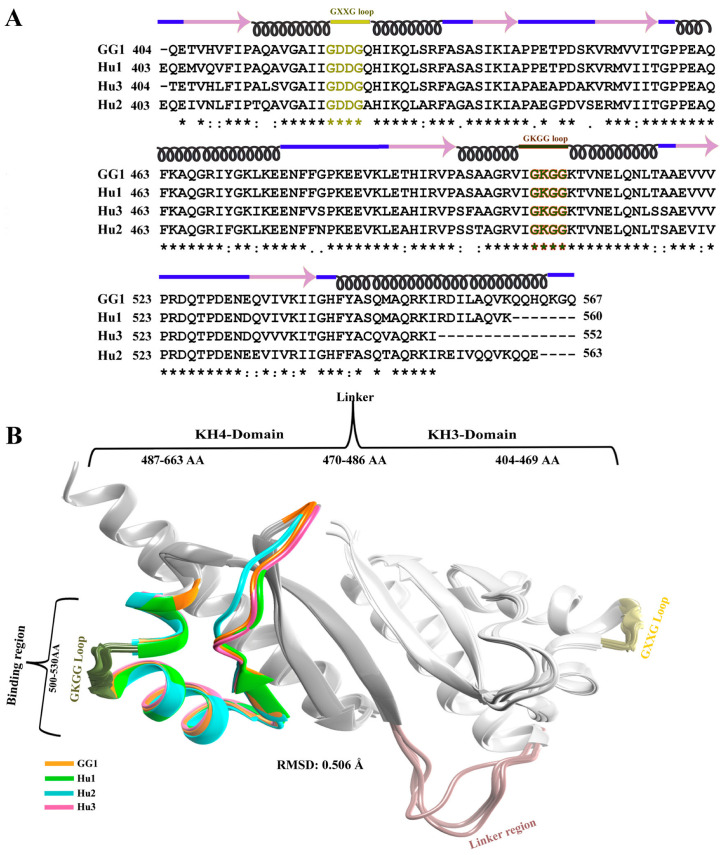
Comparative analysis of KH3-4 domains of human IGF2BPs and gallus gallus IGF2BP1. (**A**) Multiple sequence alignment of gallus gallus (GG1) IGF2BP1 and human IGF2BP1 (Hu1), IGF2BP2 (Hu2), and IGF2BP3 (Hu3)’s KH3-4 domains. The conserved motif involved in the binding is highlighted in light olive (GXXG) and green–red color (GKGG). The secondary structure is shown above the sequences. Alpha helices are indicated in black color, β-sheets in plum color, and loops in blue color. (**B**) Structural analysis of KH3-4 domains of all human IGF2BPs and gallus gallus IGF2BP1 with their respective colors. KH3 domains, linkers, and KH4 domains are represented in white, rosy brown, and dark grey colors, respectively.

**Figure 2 ijms-25-11118-f002:**
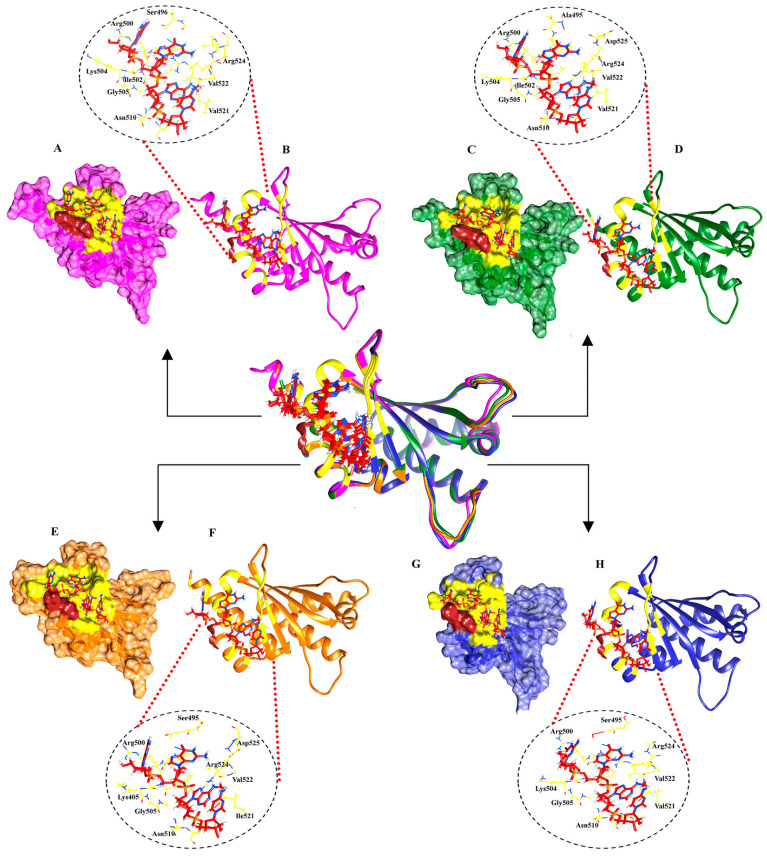
Binding pattern of GGm^6^AC RNA motif with KH3-4 domains of IGF2BPs. (**A**) Surface representation of gallus gallus KH3-4 domains (pink) with GGm^6^AC RNA (red). (**B**) The same complex is indicated using a ribbon for the protein and zoomed out for highlighting the binding residues with RNA motif. The human IGF2BP1,2 and 3 KH3-4 domains (**C**–**H**) are indicated in surface and ribbon representations with green, orange, and blue colors, respectively. In all complexes, the 503GKKG506 loop of KH4 (brown), GGm^6^AC RNA motif (red), and the binding region (yellow) are highlighted. The binding residues of KH4 domains are labeled in black color.

**Figure 3 ijms-25-11118-f003:**
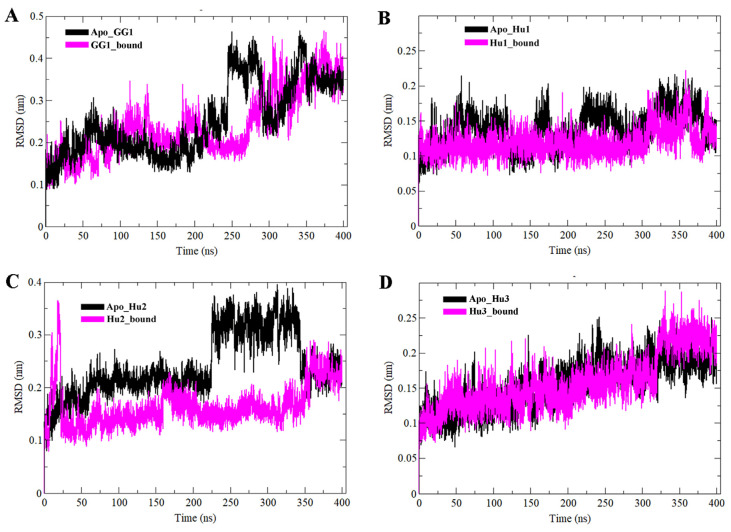
Time-dependent analysis of MD trajectories for a 400 ns time scale to investigate the stability and deviation of apo IGF2BPs (KH3-4) and their bound states. (**A**) gallus gallus Apo_GG1 and GG1_bound are illustrated in black and pink colors, respectively. RMSD plots (**B**–**D**) for human Apo_Hu1, Apo_Hu2, Apo_Hu3 and their bound states. In all complexes, apo and bound systems are represented by black and pink colors, respectively.

**Figure 4 ijms-25-11118-f004:**
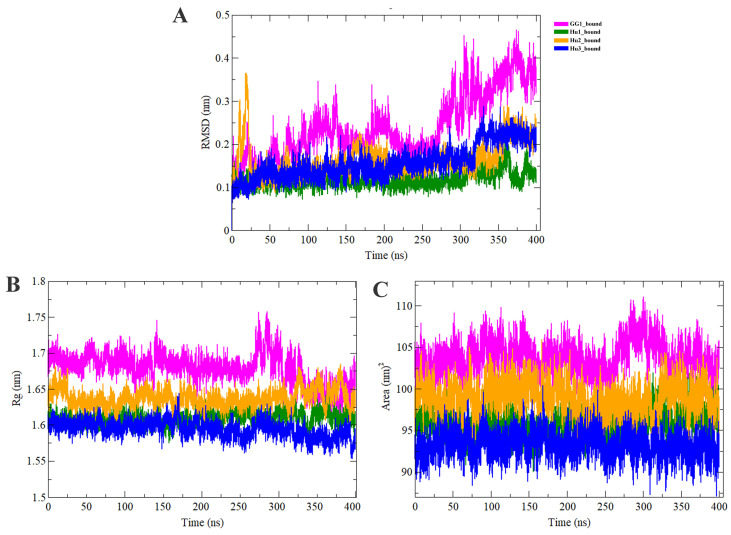
Comparative analysis of RMSD, Rg, and SASA of IGF2BPs (KH3-4) with GGm^6^AC complexes at a 400 ns MD simulation. (**A**) Root mean square deviation (RMSD) (**B**) Radius of gyration (Rg) throughout the simulation. (**C**) Solvent-accessible surface area (SASA).

**Figure 5 ijms-25-11118-f005:**
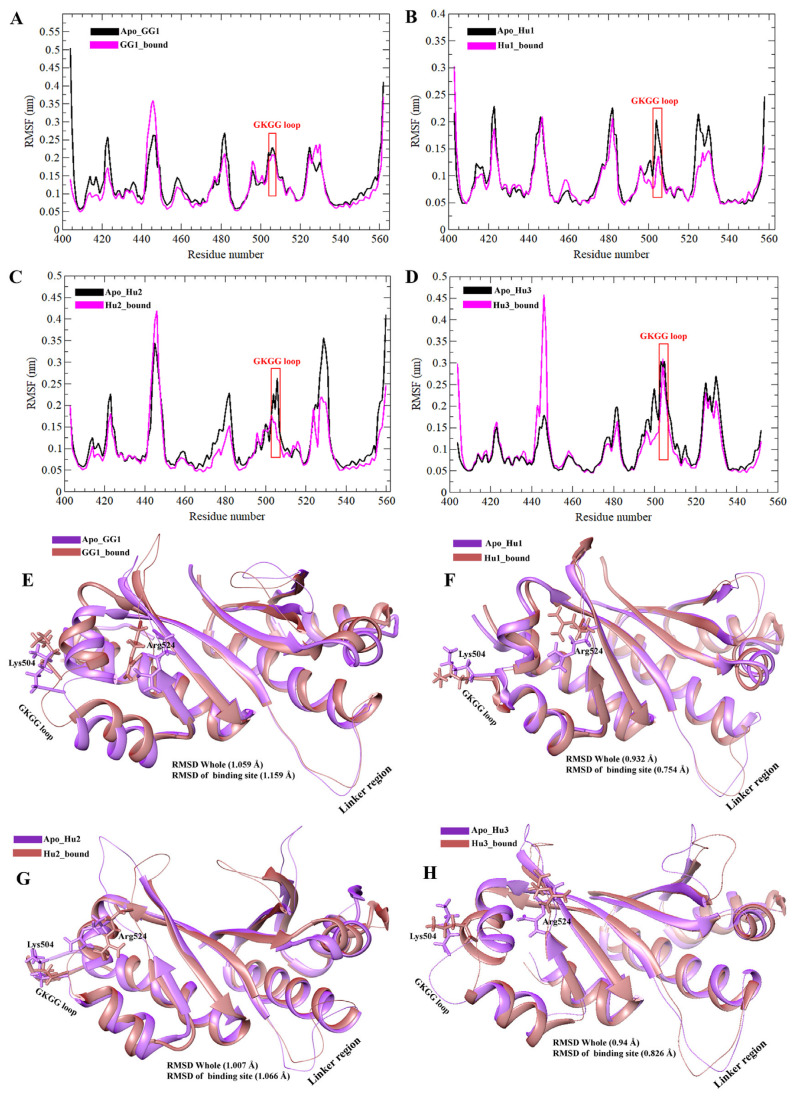
RMSF analysis and RMSD calculation by superimposition of Apo IGF2BPs (KH3-4) domains and their bound states with GGm^6^AC at 400 ns. (**A**) Comparative RMSF plots of gallus gallus Apo_GG1 and GG1_bound are illustrated in black and pink colors respectively. Similarly, RMSF plots for human Apo_Hu1, Apo_Hu2, Apo_Hu3 (**B**–**D**), and their bound states follow the same color scheme: black for apo and pink for bound. (**E**–**H**) Superimposition of 3D structures of Apo_GG1, Apo_Hu1, Apo_Hu2, and Apo_Hu3 with their respective bound complexes. Superimposed Apo and bound 3D structures are shown in purple and brown colors, respectively.

**Figure 6 ijms-25-11118-f006:**
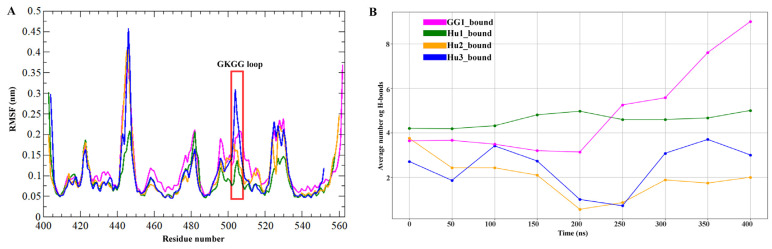
Comparative analysis of RMSF and average number of hydrogen bonds of IGF2BPs with GGm^6^AC complexes at 400 ns MD simulation. (**A**) RMSF values of alpha carbon over the entire simulation. (**B**) Average number of hydrogen bonds over the entire simulation.

**Figure 7 ijms-25-11118-f007:**
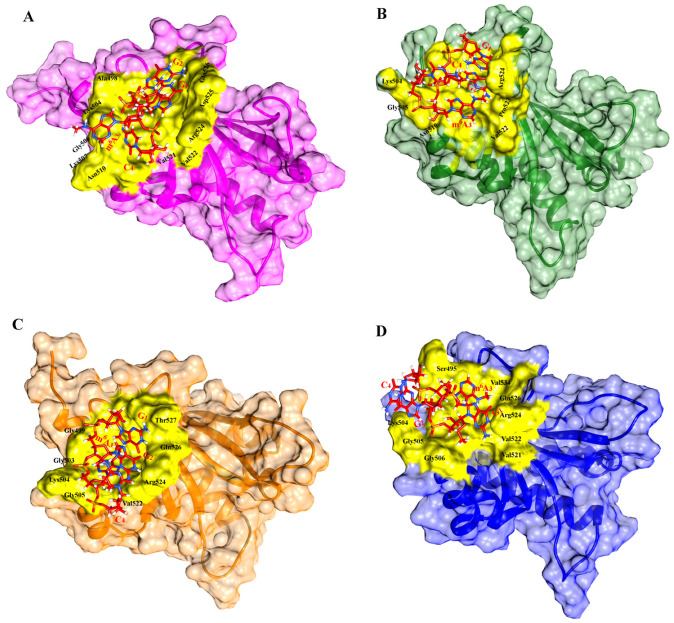
Time-dependent binding dynamics of GGm^6^AC RNA motif at IGF2BPs KH3-4 domains. (**A**) gallus gallus GG1_bound (pink) and (**B**) human Hu1_bound (green) binding with GGm^6^AC (red) at 400 ns. (**C**) Hu2_bound (orange) and (**D**) Hu3_bound (blue) at 150 and 120 ns MD simulation time scales, respectively, showed an interaction with the GGm^6^AC (red) RNA motif. The binding region is highlighted in yellow color, and some core binding residues at the groove region are labeled in black color. The GGm^6^AC RNA motif is labeled in red color in all complexes.

**Figure 8 ijms-25-11118-f008:**
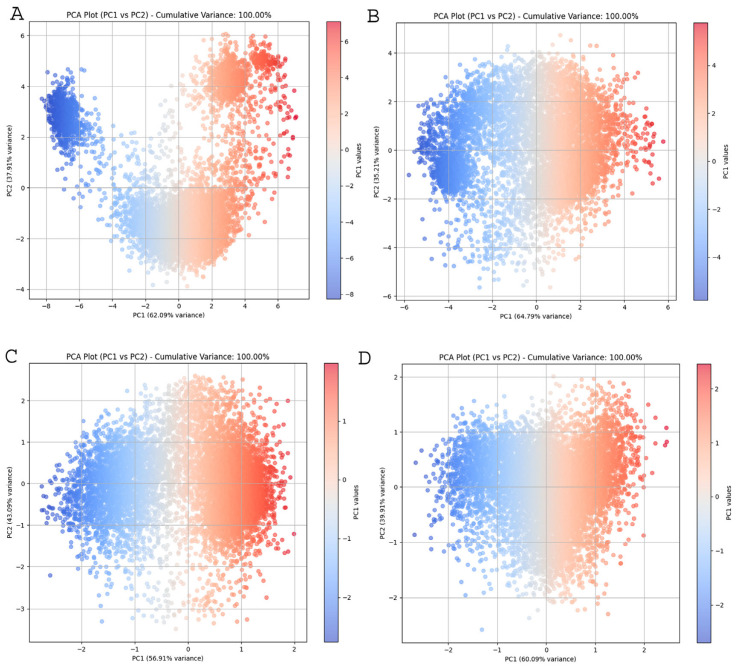
Principal component analysis 2D projection scatters plot of 400 ns MD trajectories for apo and bound IGF2BP1 (KH3-4) with GGm^6^AC. Panels (**A**) apo_GG1, (**B**) GG1_bound, (**C**) apo_Hu1, and (**D**) Hu1_bound represent 2 D plots.

**Figure 9 ijms-25-11118-f009:**
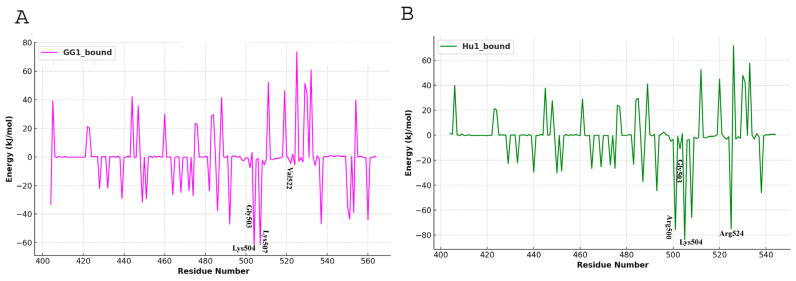
Per-residue decomposition of binding enthalpy from MD trajectories estimated by the MM/PBSA method. Binding energy decomposition at residue basis for (**A**) GG1_bound and (**B**) Hu1_bound complexes are indicated in pink and green colors, respectively.

**Figure 10 ijms-25-11118-f010:**
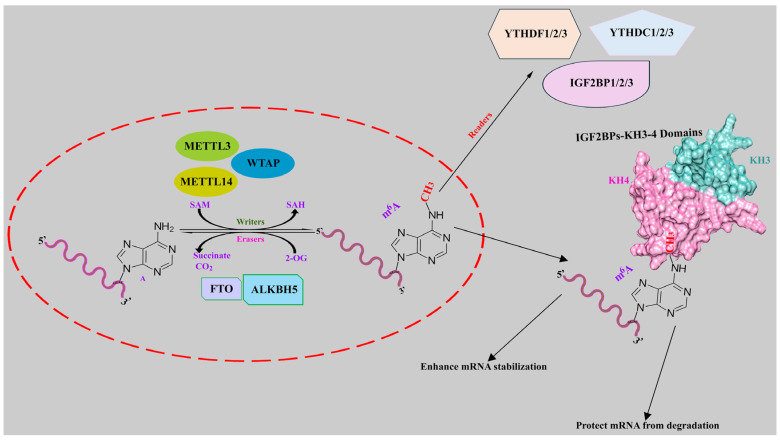
The methylation process of m^6^A in consensus motif of target transcripts. The modification of m^6^A is regulated by ‘writers’, ‘readers’, and ‘erasers’. ‘Writers’ such as METTL3, METTL14, an d WTAP regulate m^6^A methylation. RNA m^6^A demethylation is prompted by eraser proteins such as FTO and ALKBH5. IGF2BPs have a role, like other reader proteins, in reading the m^6^A binding sites of target mRNAs to protect mRNA from degradation and promote cancer proliferation.

**Table 1 ijms-25-11118-t001:** Energy contributions (KJ/mol) of gallus gallus and human IGF2BP1 (KH3-4) in complex with GGm^6^AC RNA motif.

Complex	E_vdw_	E_elec_	G_sol-polar_	Gsol_-non-polar_	ΔG_binding_
GG1_bound	−166.518 +/− 38.842 kJ/mol	−342.658 +/− 158.491 kJ/mol	450.713 +/− 181.749 kJ/mol	−20.032 +/− 4.320 kJ/mol	−78.494 +/− 71.542 kJ/mol
Hu1_bound	−179.233 +/− 28.346 kJ/mol	−493.531 +/− 99.375 kJ/mol	433.563 +/− 124.414 kJ/mol	−19.425 +/− 3.174 kJ/mol	−258.627 +/− 52.896 kJ/mol

## Data Availability

The raw data, including the docking models of IGF2BPs with the GGm^6^AC RNA motif and molecular dynamics simulation trajectories at 400 ns for all complexes, is publicly available on Zenodo [69].

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
