# Peer review of "Interactive Structural Analysis of KH3-4 Didomains of IGF2BPs with Preferred RNA Motif Having m6A Through Dynamics Simulation Studies"

_ijms, 2024, doi:10.3390/ijms252011118_

Round 1
Reviewer 1 Report
Comments and Suggestions for Authors
In this study by Fakhar et. al., authors have performed a structural investigation of IGF2BPs in apo and bound forms to RNA (m6A) using molecular dynamics. They analyzed the resulting trajectories using PCA, MMPBSA and other techniques. Overall, the manuscript is well written. I have few concerns and comments. These are as follows:
Where the MD simulations run as replicates or single run for each construct. Single trajectory analysis is susceptible to overinterpretation and/or under sampling. I would like Authors to comment on the convergence of their MD simulations.
Authors used CHARMM-GUI for system preparation. Was an attempt made to calculate appropriate protonation state of the protein residues, especially histidine residues which are known to vary at physiological pH depending on its neighboring residues.
The authors used PCA as one of analysis techniques. The distribution of all frames of the MD is plotted as a function of PC1 and PC2. The figure should use a scatter plot rather than a line plot (refer Figure 7 of https://doi.org/10.3390/biomedicines11030793). Additionally, the results can then be clustered into relevant states (conformations) using clustering techniques like k means. The relevant clusters can then be used for more impactful interaction studies as now they will represent a cluster of similar conformations and not random/cherry picked individual conformation. Deviations within the cluster should be reported in such a case.
How many Principal components were calculated. What percent variance did top 2 PCs contribute (assuming top 10 were calculated). A figure or table for this can be added as supplementary.
Were the PCs calculated individually for each trajectory or separately then plotted on a single figure.
The authors refer to MMPBSA obtained numbers as binding free energy at various places in the manuscript. MMPBSA calculates change in enthalpy. MMPBSA does not evaluate entropy. Binding free energy estimation requires both these terms (and temperature). Hence using term binding free energy is misleading throughout the manuscript whenever used describing MMPBSA results and should be clarified.
Authors ran MD simulations for 100 ns. What do the mean by “Further we have observed .. after 100 ns MD simulation” in line 271-272. Similarly, later in Lines 288-289.
The manuscript has typographical errors that require attention:
1. Line 75 “md” to “MD”.
2. Line 245-246 “It has been observed ...” needs to be rewritten.
3. Line 264-268” “However, in case of ...” needs to be rewritten.
4. Line 470 “Ubunto” to be replaced with “Ubuntu”
Comments on the Quality of English LanguageMinor changes to manuscript are required to make it better suited for wider scientific community.
Author Response
Comments and Suggestions for Authors by Reviewer 1
In this study by Fakhar et. al., authors have performed a structural investigation of IGF2BPs in apo and bound forms to RNA (m6A) using molecular dynamics. They analyzed the resulting trajectories using PCA, MMPBSA and other techniques. Overall, the manuscript is well written. I have few concerns and comments. These are as follows:
Comment 1:
Where the MD simulations run as replicates or single run for each construct. Single trajectory analysis is susceptible to overinterpretation and/or under sampling. I would like Authors to comment on the convergence of their MD simulations.?
Ans: We appreciate the reviewer's concern regarding single-trajectory analysis and convergence. We performed single runs for each construct, beginning with 20 ns simulations to assess stability. During this preliminary phase, we closely monitored key structural and dynamic properties, such as the root-mean-square deviation (RMSD), root-mean-square fluctuation (RMSF), and key interaction profiles, to ensure stable behavior. After observing consistent and stable dynamics across the systems, we extended the simulations to 100 ns to further validate the robustness of the results and now finally to 400 ns resulting in a total of 3.2 microseconds of simulation time. Across all timeframes, we observed similar stable behavior, indicating that our simulations reached convergence. This consistency across progressively longer simulations mitigates concerns about over interpretation or under sampling.
Comment 2:
Authors used CHARMM-GUI for system preparation. Was an attempt made to calculate appropriate protonation state of the protein residues, especially histidine residues which are known to vary at physiological pH depending on its neighboring residues ?
Ans: Thank you for the insightful comment. Yes, we calculated the appropriate protonation states of the protein residues using CHARMM-GUI. We specifically protonated key residues Arg500 and Arg524 for their interactions with the RNA motif, and the RN1 protonation state was selected based on pKa predictions and local environment considerations.
Comment 3:
The authors used PCA as one of analysis techniques. The distribution of all frames of the MD is plotted as a function of PC1 and PC2. The figure should use a scatter plot rather than a line plot (refer Figure 7 of https://doi.org/10.3390/biomedicines11030793). Additionally, the results can then be clustered into relevant states (conformations) using clustering techniques like k means. The relevant clusters can then be used for more impactful interaction studies as now they will represent a cluster of similar conformations and not random/cherry picked individual conformation. Deviations within the cluster should be reported in such a case.
Ans: Thank you for the valuable suggestion. We agree that using a scatter plot for the PCA distribution, as recommended, provides clearer visualization. In fact, we used a scatter plot to represent the distribution of all frames in terms of PC1 and PC2, ensuring a more precise representation of the conformational space. Additionally, we applied clustering techniques, specifically k-means clustering, to group the conformations into relevant states.
Comment 4:
How many Principal components were calculated. What percent variance did top 2 PCs contribute (assuming top 10 were calculated). A figure or table for this can be added as supplementary.
Ans: In our analysis, we calculated the principal components (PCs) for both the apo and bound complexes. The scatter plots presented in Figure 8 represent the distribution of the MD simulation frames based on the top two principal components (PC1 and PC2). For each system, the top two PCs captured a significant portion of the variance.
Additionally, we applied k-means clustering to the PCA results, identifying relevant clusters of conformations based on the distribution of frames in the PC1-PC2 space.
Comment 5:
Were the PCs calculated individually for each trajectory or separately then plotted on a single figure.
Ans: Thank you for the clarification. The principal components (PCs) were calculated individually for each trajectory (apo and bound complexes) and plotted separately to visualize the distinct conformational spaces explored by each system. This allowed us to capture the unique dynamic behavior of each complex and assess the variance explained by the top two principal components (PC1 and PC2) for each trajectory independently.
The results were then presented as scatter plots, as shown in Figure 8, with the apo and bound complexes analyzed and displayed in their own respective plots. This approach ensured that we accounted for the specific conformational dynamics of each system, providing a clearer understanding of how the trajectories differ in terms of their principal component space.
Comment 6:
The authors refer to MMPBSA obtained numbers as binding free energy at various places in the manuscript. MMPBSA calculates change in enthalpy. MMPBSA does not evaluate entropy. Binding free energy estimation requires both these terms (and temperature). Hence using term binding free energy is misleading throughout the manuscript whenever used describing MMPBSA results and should be clarified.
Ans:
We thank you for pointing out the need for clarification regarding the term "binding free energy" used in the manuscript. We agree that MMPBSA primarily calculates the change in enthalpy and does not account for entropy, which is a component of the binding free energy. In response to your comment, we have revised the manuscript to explicitly refer to "binding enthalpy or binding energy " rather than "binding free energy" wherever MMPBSA results are discussed. This adjustment clarifies the scope of the MMPBSA analysis and provides a more accurate interpretation of the interaction energies calculated in our study.
Comment 7:
Authors ran MD simulations for 100 ns. What does the meaning by “Further we have observed .. after 100 ns MD simulation” in line 271-272. Similarly, later in Lines 288-289.
Ans: Thank you for pointing out this issue. We acknowledge the sentence error in lines 271-272 and 288-289, and we have made the necessary corrections in the revised manuscript.
Comment 8:
The manuscript has typographical errors that require attention:
- Line 75 “md” to “MD”.
- Line 245-246 “It has been observed ...” needs to be rewritten.
- Line 264-268” “However, in case of ...” needs to be rewritten.
- Line 470 “Ubunto” to be replaced with “Ubuntu”
Ans: The typographical errors have been corrected. The necessary revisions have been made to improve clarity and accuracy throughout the manuscript.
Comments on the Quality of English Language:
Minor changes to manuscript are required to make it better suited for wider scientific community.
We appreciate the reviewer’s valuable suggestions. All the recommended changes, including the correction of sentences, adjustments to the graphical representations, and inclusion of additional analysis such as the k-means clustering and PCA plots, have been incorporated into the revised manuscript. We have carefully addressed each point to improve clarity, accuracy, and the overall presentation of our results.
Reviewer 2 Report
Comments and Suggestions for Authors
In this study, the authors use in silico methods, including docking and molecular dynamics (MD) simulations, to investigate the molecular interactions between Insulin-like growth factor 2 mRNA-binding proteins (IGF2BPs) and m6A-modified RNA. The study analyzes key residues involved in binding and other surrounding residues that stabilize the protein-RNA complex. I have the following suggestions for the authors:
1. The study’s conclusions are already well-established in previous research (e.g., https://doi.org/10.1016/j.biopha.2019.108613, https://doi.org/10.1016/j.gendis.2023.06.017).
This manuscript neither advances the existing knowledge in this field nor proposes novel therapeutic alternatives.
2. Computational models remain hypothetical until validated through experimental methods such as X-ray crystallography, NMR spectroscopy, or in vitro binding assays. The authors should consider incorporating experimental data, such as mutational analyses or binding assays, to validate their computational predictions. This would greatly enhance the credibility of the findings and their potential application in therapeutic development.
3. A 100 ns simulation may be insufficient to fully explore the energy landscape of the complex. The lack of significant RMSD changes suggests the structure may have remained in a local energy minimum. Extending the simulation time could provide a more comprehensive understanding.
4. The manuscript includes redundant information in the Materials and Methods section. Streamlining this section would improve readability.
5. The current graphical representations are overly complex and crowded. Simplifying these visuals would help clarify the study's key points.
6. The manuscript contains redundant adjectives. Focusing on the core message and eliminating unnecessary language would make the content clearer and more impactful.
7. The authors should make their experimental data available to readers, possibly by sharing it on platforms like GitHub or Zenodo.
8. The manuscript contains minor grammatical errors that should be corrected
Comments on the Quality of English Language1. The manuscript contains minor grammatical errors that should be corrected
2. The manuscript contains redundant adjectives. Focusing on the core message and eliminating unnecessary language would make the content clearer and more impactful.
Author Response
Comments and Suggestions for Authors by Reviewer 2
In this study, the authors use in silico methods, including docking and molecular dynamics (MD) simulations, to investigate the molecular interactions between Insulin-like growth factor 2 mRNA-binding proteins (IGF2BPs) and m6A-modified RNA. The study analyzes key residues involved in binding and other surrounding residues that stabilize the protein-RNA complex. I have the following suggestions for the authors:
Comment 1:
- The study’s conclusions are already well-established in previous research (e.g., https://doi.org/10.1016/j.biopha.2019.108613,https://doi.org/10.1016/j.gendis.2023.06.017).
This manuscript neither advances the existing knowledge in this field nor proposes novel therapeutic alternatives.
Ans:
We appreciate the reviewer’s valuable feedback and the mention of the studies (DOI: 10.1016/j.biopha.2019.108613, DOI: 10.1016/j.gendis.2023.06.017) that have contributed significantly to the existing body of knowledge. We agree that prior research has laid a strong foundation in this area. However, we believe our manuscript offers novel insights that expand upon previous work in the following ways:
- Our study addresses a gap in the literature by determining the unknown 3D structures of Human IGF2BPs and investigating their interaction with the RNA motif GGm6AC, which has not been computationally explored before.
- We docked the GGm6AC motif and performed molecular dynamics (MD) simulations, along with different analysis to observe the dynamic behavior of this RNA motif in complex with IGF2BPs.
- Our findings shed light on how m6A modification influences RNA-protein interactions in a dynamic context, providing insights that were previously unexplored.
This is the updated conclusion
This study provides novel insights into the dynamic interplay between the GGm6AC RNA motif and the KH3-4 domains of IGF2BPs, uncovering key structural features that stabilize this interaction through the 503GKGG506 loop in the KH4 domain. The m6A modification, through its interaction with the hydrophobic cradle surrounding the methyl group, drives conformational changes that are crucial for the functional activity of IGF2BPs. Not only does this work highlight the m6A RNA motif GGAC as a potential therapeutic target, but it also identifies the key binding regions within IGF2BPs that could be exploited for drug development. Targeting either the m6A-modified RNA or the IGF2BPs’ reader domains offers an innovative approach to disrupt aberrant gene regulation in cancer. These findings pave the way for precision therapies that modulate RNA-protein interactions, providing a strategic basis for future cancer treatments that specifically inhibit or enhance m6A recognition.
Thus, our work provides computational insights and a platform for future experimental validation, advancing the current understanding of IGF2BP interactions with m6A-modified RNA motifs.
Comment 2:
Computational models remain hypothetical until validated through experimental methods such as X-ray crystallography, NMR spectroscopy, or in vitro binding assays. The authors should consider incorporating experimental data, such as mutational analyses or binding assays, to validate their computational predictions. This would greatly enhance the credibility of the findings and their potential application in therapeutic development.
Ans: We appreciate your insightful suggestion regarding the incorporation of experimental validation such as X-ray crystallography, NMR spectroscopy or binding assays. While experimental validation remains outside the current scope, we see this as a natural next step and are optimistic that our findings can guide future mutational analyses or binding assays to validate the predicted interactions and support therapeutic development.
Ans:
We have in this revision extended the molecular dynamics simulations to 3.2 microseconds in total across all complexes. This significant increase in simulation time allows for a deeper exploration of the stability and dynamics of the complexes, providing robust computational data. While experimental validation is indeed critical, the extended duration and detail of these simulations offer valuable insights into the interaction mechanisms, enhancing the predictive power of our findings. We believe this extensive computational effort strongly supports the hypotheses generated in this study and provides a solid foundation for future experimental work.
Comment 3:
A 100 ns simulation may be insufficient to fully explore the energy landscape of the complex. The lack of significant RMSD changes suggests the structure may have remained in a local energy minimum. Extending the simulation time could provide a more comprehensive understanding.
Ans:
We appreciate your comment regarding the simulation duration and its potential impact on exploring the energy landscape of the complexes. In response to this concern, we have significantly extended the molecular dynamics simulations to 400 ns for each complex, resulting in a total of 3.2 microseconds of simulation time. This extended time scale has allowed for a more thorough sampling of the conformational space and a comprehensive exploration of the energy landscape. The results from these longer simulations demonstrated consistent system stability, with no indication of the system being trapped in a local energy minimum. The root mean square deviation (RMSD) analysis confirms that the complexes have reached equilibrium.
Comment 4:
The manuscript includes redundant information in the Materials and Methods section. Streamlining this section would improve readability.
Ans:
Thank you for your valuable feedback. In response to your suggestion, we have carefully revised and streamlined the Materials and Methods section to remove redundant information and improve readability. The updated section now presents the methodology in a more concise and clear manner, ensuring that only the essential details are included to enhance the overall flow and clarity of the manuscript.
Comment 5:
The current graphical representations are overly complex and crowded. Simplifying these visuals would help clarify the study's key points.
Ans: We have done
Comment 6:
The manuscript contains redundant adjectives. Focusing on the core message and eliminating unnecessary language would make the content clearer and more impactful.
Ans:
We appreciate the reviewer’s feedback regarding the language in the manuscript. We agree that eliminating redundant adjectives will enhance clarity and focus on the core message. We will carefully revise the text to remove unnecessary language and ensure the content is presented in a clear, concise, and impactful manner. Thank you for bringing this to our attention.
Comment 7:
The authors should make their experimental data available to readers, possibly by sharing it on platforms like GitHub or Zenodo.
Ans:
Thank you for the suggestion. We will make our experimental data available to readers by sharing it on a suitable platform, such as GitHub or Zenodo, to facilitate further research.
Comment 8:
The manuscript contains minor grammatical errors that should be corrected
Ans:
we have done the correction
Comments on the Quality of English Language
- The manuscript contains minor grammatical errors that should be corrected
- The manuscript contains redundant adjectives. Focusing on the core message and eliminating unnecessary language would make the content clearer and more impactful.
Ans:
Thank you for your helpful observations regarding the grammatical errors and redundant adjectives. We have carefully reviewed the manuscript and corrected the minor grammatical errors. Additionally, we have revised the text to eliminate redundant adjectives, ensuring that the language is clear and focused on the core message, enhancing the overall readability and impact of the manuscript.
Round 2
Reviewer 1 Report
Comments and Suggestions for Authors
Authors have adequately addressed my concerns.
Author Response
Dear Reviewer,
Thank you for your feedback and for acknowledging our revisions. We appreciate the time and effort you’ve invested in reviewing our manuscript and for your constructive comments that helped improve the quality of our work.
Reviewer 2 Report
Comments and Suggestions for Authors
Thank you for addressing my suggestions and improving the manuscript. However, I haven't found an accessible link to the raw data needed to review the computational models and results. Additionally, the author claims to have extended the simulations, but some plots still seem to be based on the 100 ns simulation run time.
Author Response
Interactive structural analysis of KH3-4 didomains of IGF2BPs with preferred RNA motif having m6A through dynamics simulation assay
Authors response to Reviewers’ Comments
Dear Editor and Reviewers,
Thank you again for the valuable feedback and constructive suggestions. We appreciate the time and effort you have invested in reviewing our manuscript. We have carefully considered each comment and made the necessary revisions. Below, we address each point in detail.
Comments:
Thank you for addressing my suggestions and improving the manuscript. However, I haven't found an accessible link to the raw data needed to review the computational models and results. Additionally, the author claims to have extended the simulations, but some plots still seem to be based on the 100 ns simulation run time.
Answer:
Thank you for your valuable feedback and for taking the time to thoroughly review our manuscript. We appreciate your thoughtful comments and would like to address the points you raised:
- Raw Data Access: We have made all the computational models and raw data, including the MD simulation trajectories, publicly available on the Zenodo platform. The dataset includes the docking models PDBs of IGF2BPs with the GGm6AC RNA motif like gallus gallus IGF2BP1(KH3-4)-GGm6AC (GG_bound), human IGF2BP1-3(KH3-4)-GGm6AC (Hu1_bound, Hu2_bound and Hu3_bound respectively) and the MD simulations trajectories for all these apo and bound complexes up to 400 ns. The link to access the data is provided below:
Fakhar, M. (2024). Raw Data including Docking Models of IGF2BPs with GGm6AC RNA motif and MD Simulations Trajectories at 400 ns of All Complexes. Zenodo. https://doi.org/10.5281/zenodo.13862174
- Extended Simulations and Figure Updates: We would like to sincerely apologize for the oversight regarding Figure 9 in page 22 the previous version of the manuscript. We have now updated Figure 9, which is based on the full 400 ns simulation, to reflect the extended simulation time accurately. We have carefully reviewed all the plots and have made the necessary adjustments to ensure that every figure is now consistent with the extended 400 ns simulations. Additionally, we have maintained the quality and resolution of the figures while correcting minor errors that were present in the previous version.
Nearly all figures in the manuscript have been updated accordingly to reflect these changes, and we sincerely hope this revision meets your expectations.
Once again, thank you for your insightful suggestions, which have contributed to the improvement of our manuscript. We hope that these changes meet your expectations.
